

# Retrievals of Aerosol Optical and Microphysical Properties from Imaging Polar Nephelometer Scattering Measurements

W. Reed Espinosa[1,2], Lorraine Remer[1,2], Oleg Dubovik[3], Luke Ziemba[4], Andreas Beyersdorf[4,5], F. Daniel Orozco[1,2], Gregory Schuster[4], Tatyana Lapyonok[3], David Fuertes[6], and J. Vanderlei Martins[1,2]

[1]Department of Physics, University of Maryland Baltimore County, 1000 Hilltop Circle, Baltimore, MD 21250, USA
[2]Joint Center for Earth Systems Technology, University of Maryland Baltimore County, 5523 Research Park DR, Baltimore, MD 21228, USA
[3]Laboratoire d'Optique Atmosphérique, UMR8518, CNRS - Université de Lille 1, 59655, Villeneuve d'Ascq, France
[4]Langley Research Center Science Directorate, National Aeronautics and Space Administration, Hampton, Virginia, USA
[5]Department of Chemistry and Biochemistry, California State University San Bernardino, 5500 University Parkway, San Bernardino, CA 92407, USA
[6]GRASP-SAS, Bat-P5, Université de Lille 1, 59655, Villeneuve d'Ascq, France

*Correspondence to:* W. Reed Espinosa (reedespinosa@umbc.edu)

**Abstract.** A method for the retrieval of optical and microphysical properties from in situ light scattering measurements is presented and the results are compared with existing measurement techniques. The Generalized Retrieval of Aerosol and Surface Properties (GRASP) is applied to airborne and laboratory measurements made by a novel polar nephelometer. This instrument, the Polarized Imaging Nephelometer (PI-Neph), is capable of making high accuracy field measurements of phase function and

degree of linear polarization, at three visible wavelengths, over a wide angular range of $3°$ to $177°$ . The resulting retrieval produces particle size distributions (PSD) that agree, to within experimental error, with measurements made by commercial optical particle counters (OPCs). Additionally, the retrieved real part of the refractive index is generally found to be within the predicted error of 0.02 from the expected values for three species of humidified salts particles, whose refractive index is well established. The airborne measurements used in this work were made aboard the NASA DC-8 aircraft during the Studies

of Emissions and Atmospheric Composition, Clouds and Climate Coupling by Regional Surveys (SEAC[4]RS) field campaign, and the inversion of this data represent the first aerosol retrievals of airborne polar nephelometer data. The results provide confidence in the refractive index product, as well as in the retrieval's ability to accurately determine PSD, without assumptions about refractive index that are required by the majority of OPCs.

## 1 Introduction

Aerosols, and their interaction with clouds, play a key role in the climate of our planet. Additionally, measurements of aerosols are crucial to a wide range of direct applications, ranging from the monitoring of clean rooms to the impact of air quality on public health. Despite the importance of these particles, accurate in situ measurements of their optical and microphysical properties have remained a significant challenge.



Optical techniques of particle sizing typically capitalize on the approximately monotonic increase in the amount of light scattered by a single particle as a function particle size. These instruments are among the most widespread and precise available, but the vast majority of optical particle counter (OPC) designs require significant assumptions about the aerosol being sampled. These simplifications result from the limited information content present in typical OPC measurements, which frequently

sample scattered light over a single angular range, often 4 to 22 degrees (Pinnick et al., 2000) or roughly 30 to 150 degrees (Cai et al., 2008) in so called wide angle OPCs. These assumptions, generally regarding real refractive index, absorption and particle morphology can lead to significant biases in the resulting particle size distributions (PSD) and generally constitute the bulk of the measurement error (Pinnick et al., 2000). Additionally, in situ measurements of many of these characteristics, like aerosol refractive index or particle sphericity for example, are still virtually nonexistent, especially at altitudes far from the

surface.

A less common approach to characterizing particles is through polar nephelometer measurements of light scattering from an ensemble of particles over a large number of angular regions. This approach provides a large amount of information about the sample reducing the total number of assumptions required and the resulting biases in the retrieved products. Unfortunately, deploying field instruments with these capabilities can be quite challenging, and airborne measurements of common aerosols

using this technique have previously been unavailable. Additionally, the inversion of multi-angular data, is significantly more complex than the inversion of light scattering intensity over a single angular range.

In spite of the complexities associated with multi-angle measurements and the corresponding inversions, there have been several successful attempts over the past four decades to retrieve particle properties from polar nephelometer data. The first published inversion of this kind was made by Eiden in 1966, who used multi-wavelength polarization data to retrieve the

20 complex refractive index of an ambient aerosol, as well as match one of three predefined aerosol PSD models (Eiden, 1966). Jones et al. (1994) used intensity measurements to size monodisperse, polystyrene latex (PSL) spheres, as well as determine their complex index of refraction. Intensity and polarization measurements of ambient aerosols made by the Tohoku University single wavelength polar nephelometer in Sendai, Japan have been inverted to obtain complex refractive index and number concentrations in six log spaced size bins (Tanaka et al., 1983; Zhao, 1999). There have also been attempts to retrieve only the

25 refractive index, while constraining the model's size distribution with a traditional particle sizer (Barkey et al., 2007, 2010). The converse approach was reported by Lienert et al. (2003), who took polarized measurements of sea spray and determined PSD by assuming a refractive index value expected for sodium chloride particles at the ambient relative humidity. Most recently, Sviridenkov et al. (2014) obtained both complex refractive index and PSD from three wavelength intensity measurements made with a commercially available polar nephelometer. All of these retrieval efforts have assumed spherical particles, and all

30 measurements were made in the visible spectrum, except in the case of Jones et al. (1994) who used measurements made in the near-infrared. The only polar nephelometer retrievals to incorporate a non-spherical component in the scattering model were performed by Dubovik et al. (2006), who fit laboratory measurements of desert dust.

In this work we apply a complex inversion algorithm, specifically the Generalized Retrieval of Aerosol and Surface Properties (GRASP), to airborne and laboratory measurements made with the Polarized Imaging Nephelometer (PI-Neph), a multi-

35 wavelength, multi-angle light scattering instrument. The GRASP retrieval makes no assumptions about the number of modes in





the size distribution or the complex refractive index, and it allows for both spherical and spheroidal scatterers. This represents a significant increase in complexity when compared to previous in situ scattering inversions. In addition to the generality of the retrieval, this work represents the first time that any aerosol retrieval algorithm has been applied to airborne polar nephelometer measurements. Furthermore, the ambient airborne measurements presented here were made in parallel to a large variety of

independent instrumentation, allowing for very robust inter-comparisons of the retrieved products.

## 2   Inversion methodology

Aerosol scattering matrix elements are measured in situ with a polar nephelometer and feed into a microphysical retrieval algorithm in order to obtain aerosol size distribution, complex refractive index (m) and a percentage of spherical particles. These measurements include a combination of artificially suspended laboratory data as well as airborne data taken over the

continental United States during the Studies of Emissions and Atmospheric Composition, Clouds and Climate Coupling by Regional Surveys (SEAC[4]RS) field experiment in 2013. GRASP, a versatile open source software package (http://www.grasp-open.com) capable of performing inversions on a wide variety of atmospheric optical measurements, was used to obtain the retrieved microphysical parameters. A detailed description of the GRASP retrieval algorithm and its capabilities can be found in Dubovik et al. (2011, 2014).

### 2.1   Polarized Imaging Nephelometer

In an effort to advance in situ characterization of atmospheric aerosols, the Laboratory for Aerosols, Clouds and Optics (LACO) at the University of Maryland, Baltimore County (UMBC) has developed a novel instrument concept called the Imaging Nephelometer (Dolgos and Martins, 2014). This compact device is capable of measuring scattering matrix elements with an angular resolution and range that has previously been unavailable in polar nephelometers. The imaging nephelometer design,

first realized in the PI-Neph, uses a wide field of view charge coupled device (CCD) camera to image the light scattered by aerosols in the path of a high-powered continuous wave laser. This setup capitalizes on multiple scattering locations inside the sampled volume to allow for measurements of scattering matrix elements over a very wide angular range with an angular resolution that is limited only by the number of pixels contained on the CCD. It also permits the construction of an instrument that is compact and stable enough to be flown on a variety of airborne platforms.

The aerosol sample inside the PI-Neph is illuminated sequentially by a three wavelength laser system operating at 473nm, 532nm and 671nm, as shown in Figure 1. The three beams are aligned by a system of dichroics and mirrors before having their polarization state precisely aligned by a Glan-Taylor linear polarizer. A liquid crystal variable retarder (LCVR) and Fresnel Rhomb are then used to actively rotate the polarization state of laser light. After exiting the rhomb the beam is guided, by a series of mirrors and then through a window, into a 10 liter, sealed chamber containing the aerosol sample. The laser light

traverses the length of the chamber before a corner cube retroreflector redirects the beam back toward a beam trap adjacent to the entry window. The scattered light generated by the aerosol and surrounding gas is then imaged twice by the CCD camera, once for each of two roughly orthogonal linear polarization states of the laser.



If the scattering medium is assumed to be macroscopically isotropic and symmetric then $P_{13}$ and $P_{14}$ do not contribute to the total scattered signal and the resulting pair of image intensities depend only on the product of the sample's total scattering coefficient with the first two scattering matrix elements. The images can then be processed in a manner that allows for direct measurements of $\beta_{scat}P_{11}(\theta)$ and $\beta_{scat}P_{12}(\theta)$, where $\theta$ is the zenith scattering angle (azimuthal symmetry is implied by the assumption of a macroscopically isotopic and symmetric medium). These measurements are generally made over an angular range of $3°$ to $177°$ in zenith scattering angle, with an angular resolution less than one degree. The final products are then reported at standard temperature and pressure, with the Rayleigh scattering contribution from air subtracted. In this paper normalized phase functions are defined such that the total sine weighted integral of $P_{11}(\theta)$ over all solid angles is equal to $4\pi$, with the nearest neighbor technique used to extrapolate the truncated regions around $\theta = 0°$ and $\theta = 180°$.

PI-Neph measurements have been validated by a variety of methods since the instrument's completion in the summer of 2011. Measurements of monodisperse PSL spheres have yielded results that are in excellent agreement with Mie theory, while scattering coefficient measurements made in parallel with commercially available integrating nephelometers have agreed to within 5%. A detailed summary of PI-Neph design, calibration and validation can be found in Dolgos and Martins (2014).

## 2.2 Artificial aerosol generation

Ammonium sulfate ($(NH_4)_2SO_4$), ammonium nitrate ($NH_4NO_3$), and Sodium Chloride ($NaCl$) were suspended and humidified in a laboratory setting. Figure 2 displays a schematic diagram of the particle generation and measurement setup. The salts were diluted with distilled water to a concentration of $5\text{gl}^{-1}$ before being agitated with an ultrasonic vibrator and later suspended using a single jet atomizer (TSI, model 9302). The generated aerosol was diluted with filtered compressed air before being fed into a dryer and then a humidifier. In the first stage, generated particles are dried, without heating, to a relative humidity less than 30% using a Perma Pure Nafion dryer. The dry particles are later humidified to a RH>80% using two Perma Pure Nafion humidifiers (Nafion1 and Nafion2). The humidifier and dryer consist of a Nafion membrane tubing that transfers moisture to or from the surrounding medium. The drier uses compressed air while the air passing through Nafion 1 is humidified by flowing water and then used to humidify the sample passing though Nafion 2. The separation of liquid water from the Nafion tube in contact with the aerosol sample allows for subtler control of the final relative humidity (Orozco et al., 2016). Angular scattering measurements of the aerosol were then made by the PI-Neph before the sample was discharged from the system.

The humidification system was set to relative humidity values above the deliquescence points of each salt solution, typically to an RH just over 80%. The humidity was continuously monitored throughout the measurement using RH sensors located at the PI-Neph's inlet, measurement chamber, and outlet. The stability and reproducibility of the particle generation was independently validated by the proper observation of deliquescence of different salts using an integrating nephelometer (model 3563, TSI Inc., St. Paul, MN, USA).

This setup was also used to suspend 903nm diameter monodisperse PSL spheres (Nanosphere 3900A, ThermoFisher Scientific, Fremont, CA, USA) and scattering measurements of these spheres were made by the PI-Neph at low relative humidities (RH < 20%). These measurements provide an opportunity to test the retrieval technique on an aerosol with a monodisperse size





distribution and a refractive index that is very well characterized. The PSL generation and measurements also allowed for a small, sub-degree re-alignment of the PI-Neph scattering angle calibration in the case of the salt measurements. This correction was not applied to the data used in the PSL retrievals to avoid biasing the result.

## 2.3 Ambient measurements

In addition to the laboratory measurements, inversions were performed on airborne data from the SEAC[4]RS experiment. SEAC[4]RS was a large field mission, that took place primarily over the continental United States, in August and September of 2013. Over the course of the experiment three aircraft flew 54 different instruments on a total of 57 flights in an effort to understand a broad range of atmospheric phenomenon. A detailed description of the scientific goals, aircraft and instrumentation, as well as the corresponding implementation can be found in Toon et al. (2016).

The PI-Neph made measurements aboard the NASA DC-8 aircraft during SEAC[4]RS. Ambient air was provided to the instrument through the NASA Langley Aerosol Research Group Experiment's (LARGE) shrouded diffuser inlet (McNaughton et al., 2007), which sampled isokinetically. A flow of 20 liters per minute was maintained through the PI-Neph's 10 liter sample chamber, leading to an aerosol exchange time on the order of 30 seconds. The raw sampling rate of the instrument was synchronized to match this interval, but the retrievals in this work are generally performed on time averages taken over a period

of several minutes. The sample was conditioned with a temperature-controlled drier that heated the incoming ambient air to a temperature of $35°C$ and, in almost all cases, kept the relative humidity of the sample below 40%.

   In addition to PI-Neph scattering measurements, the LARGE group made comprehensive in situ measurements of aerosol properties in parallel to the PI-Neph. These measurements, containing data on particle number density, size distribution and optical properties, are a valuable resource for the inter-comparison of PI-Neph measurements and the corresponding retrieved

microphysical properties. In this work PI-Neph retrieved size distributions will be compared extensively to measurements made by two dedicated optical particle size spectrometers (LAS model 3340, TSI Inc., St. Paul, MN, USA and model UHSAS, Droplet Measurement Technologies, Boulder, CO, USA) as well as an aerodynamic particle sizer (APS model 3321, TSI Inc., St. Paul, MN, USA). The two optical particle spectrometers also measured at low relative humidities during SEAC[4]RS, but their sample was conditioned through a drier. This approach minimizes the evaporation of volatile compounds but can also lead

to size dependent losses in the aerosol when the instrument requires relatively large flow rates, as is the case for the PI-Neph. The aerodynamic particle sizer measurements were made at ambient humidities, but the ambient RH was determined to be less than 40% in all cases shown here so differences in PSD resulting from hygroscopic growth are not expected.

   Fifty separate sampling periods, occurring over the course of ten different flights, are highlighted in this work. The flights selected represent the ten days with the highest quality PI-Neph data, for which data is available for at least one of LARGE's

dedicated particle sizers. The intervals containing the highest aerosol scattering levels during these flights were identified and a robust averaging procedure (Beaton and Tukey, 1974) was applied to periods for which no detectable changes in the normalized angular scattering data was observed. The total scattering for these averages ranged from $30 \text{Mm}^{-1}$ to just over $500 \text{Mm}^{-1}$, with a median value of $90 \text{Mm}^{-1}$. The resulting dataset represents a wide range of aerosols, including urban pollution, organics,





Saharan dust and over a dozen cases dominated by biomass burning (BB) emissions with transport ages ranging from hours to several days.

Additionally, three individual case studies were selected to provide detailed examples of PI-Neph measurements, the corresponding GRASP fit and the resulting retrieved size distributions. Two of these cases come from periods where the scattering signal was dominated by forest fire emissions, and were chosen to emphasize the subtle distinctions in angular scattering patterns that can occur, even between two aerosols of similar type. The third case consists of boundary layer (BL) measurements made over a heavily forested region of south east Missouri. This case represents one of only a couple of periods in which a significant coarse mode was observed. The sampling locations of these three cases, as well as the flight paths for the ten selected flights, are shown in Figure 3.

## 2.4 Implementation of GRASP retrieval

GRASP is a versatile software package capable of retrieving a wide range of atmospheric and surface properties from a variety of datasets. The GRASP algorithm and corresponding software builds on the successful heritage of the PARASOL (Dubovik et al., 2011), AERONET (Dubovik and King, 2000) and laboratory (Dubovik et al., 2006) retrievals.

GRASP's base aerosol model contains very few assumptions in comparison with traditional in situ or remote sensing retrieval algorithms. It includes all necessary components required to simulate a diverse range of atmospheric observations, including remote sensing (both suborbital and space-based), optical in situ and laboratory measurements. The settings of the retrieved characteristics can be flexibly adjusted to match the particular application. For example, aerosol size distribution can be represented as a superposition of several log-normal functions or as a binned continuous function with different size resolutions (it is defined in nodal points).

As an inversion concept GRASP implements Multi-Term Least Square fitting (Dubovik, 2004). This approach allows for convenient combining of different types of observations and multiple a priori constraints in a single inversion. For example, following this concept the AERONET retrieval (Dubovik and King, 2000) retrieves many parameters simultaneously: aerosol size distribution, spectral complex refractive index and fraction of spherical particles. A priori constraints on all functions (size distribution and all spectral dependencies) are assumed smooth, while a priori estimates of values are also used for some parameters. Moreover, using the same strategy a statistically optimized multi-pixel retrieval concept was realized as an option in GRASP (Dubovik et al., 2011). This concept uses additional a priori knowledge about time and space variability of the retrieved parameters in the inversion of coordinated observations (i.e. satellite observations in different pixels).

The flexibility built into the design of GRASP allows the user to select the assumptions that best match the information content of a particular dataset. Moreover, while all of the above features have never been used in one single application, they often provide important potential for evolution of each application, for example via implementing synergy retrievals using a combination of different observations. The GRASP algorithm has previously been successfully applied to both satellite and ground-based upward-looking sky radiance measurements (Dubovik et al., 2011, 2014; Xu et al., 2016), while this paper represents the first application of GRASP to polar nephelometer data.





In this work GRASP size distributions were modeled with 16 logarithmically spaced size bins, generally ranging from 50nm to 2.94μm in radius. The lower end of this range corresponds to the sensitivity limit of ensemble type, light scattering measurements, given realistic particle size distributions. The upper bound was chosen to include the vast majority of coarse mode particles capable of passing through the LARGE inlet, which has a 50% passing efficiency at an aerodynamic radius of 1.8μm (McNaughton et al., 2007). This size range was reduced to radii between 425nm and 476nm in the case of the PSL spheres, in order to better capture the fine structure of their very narrow size distribution. In all retrievals the shape of the size distribution is only constrained by a smoothness parameter and no assumptions about the number of modes are made.

The search space for the real part of the refractive index (n) is semi-continuous between 1.33 and 1.68, while the imaginary part (k) can range from 0 to $10^{-1}$. The refractive index is held constant with respect to size but is allowed to vary as a function of wavelength. GRASP's aerosol model assumes a mixture of spheres and spheroids, whose axis ratio distribution is fixed and derived from feldspar measurements made by Volten et al. (2001). It can be shown that small deviations in the spheroid component's axis ratio distribution produces negligible changes in the angular dependence of the scattered light (Dubovik et al., 2006). It is therefore believed that this fixed shape distribution is capable of accurately modeling a wide range of non-spherical aerosols. The spheroid component was omitted from the PSL retrievals due the computational demands associated with generating the required precomputed kernels for the finer size parameter grid spacing.

## 3 Retrieval results and discussion

### 3.1 Measured data and retrieval fit

In both the 50 selected SEAC[4]RS cases and in the laboratory measurements, the residuals between the GRASP fits and the PI-Neph measured values are generally within the PI-Neph instrumental error. Figure 4 shows the measured and fit $P_{11}$ and $-P_{12}/P_{11}$ for the ammonium sulfate case, and is typical of the bulk of the retrievals performed in this work. The residuals are also plotted to clearly emphasize the differences between the measurement and fit relative to the instrument's $2\sigma$ error. In the case of the $P_{11}$ data the distances between the fit and measured values are reported as:

$$RES_{P_{11}} = Log_{10}(P_{11}^{MEAS}) - Log_{10}(P_{11}^{FIT}) \tag{1}$$

with the PI-Neph error transformed accordingly. This transformation provides a measure of relative (as opposed to absolute) error, and provides a consistently sized metric across the two orders of magnitude covered by $P_{11}$. The separation in $-P_{12}/P_{11}$ data is represented simply as the difference between the measured and fit values.

$$RES_{P_{12}/P_{11}} = \left(\frac{P_{12}}{P_{11}}\right)^{FIT} - \left(\frac{P_{12}}{P_{11}}\right)^{MEAS} \tag{2}$$

Figure 5 shows the normalized scattering matrix elements at 532nm for the three selected SEAC[4]RS case studies. A strong forward peak can be seen in the forest boundary layer measurements, which is in accordance with the significant coarse mode observed by the aerodynamic and optical particle sizers. The two biomass burning cases display very similar $P_{11}$ values,





with the only significant difference being slightly enhanced forward and backward scattering in BB Plume #2. These subtle differences are likely driven by the slightly larger coarse mode present in the latter case. In contrast to $P_{11}$, $-P_{12}/P_{11}$ shows significant differences between the two biomass burning cases. The reduced magnitude of $-P_{12}/P_{11}$ in BB Plume #1 is likely driven primarily by differences in real refractive index between the two samples. This hypothesis is supported by simulations

with a Mie code (Mishchenko, Michael I and Travis, Larry D and Lacis, 2002), which demonstrated that, in the relevant size regime, changes in refractive index on the order of 0.03 had little effect on $P_{11}$ but could easily change the ratio of $P_{12}$ to $P_{11}$ by 20% or more. It is this effect, in combination with the small median size of the fine mode, that produces the highest degree of linear polarization of the three samples in the forested boundary layer case.

The spectral dependence of $\beta_{scat}P_{11}$ and $-P_{12}/P_{11}$ for the biomass burning case study sampled on August 19[th] is shown

in Figure 6. The absolute phase function values are shown here to emphasize the additional information present in the spectral dependence of the scattering intensities. It should be noted that there is also significant spectral dependence in the shape of the phase matrix elements, particularly in $-P_{12}/P_{11}$. These difference are driven primarily by changes in size parameter, but also result in some part from a non-zero spectral dependence of the complex refractive index.

The oscillations occasionally present in the data over angular scales of roughly ten degrees are likely non-physical, and ar-

15 tifacts of insufficient sampling statistics in the coarse mode. The extended spatial extent inherent to the imaging nephelometer measurement makes it especially susceptible to sampling statistic artifacts that are produced by the largest particles. These particles make up a very small fraction of the total number concentration, while simultaneously accounting for a disproportionately large portion of the total scattered light. This is especially apparent in the measurements of $-P_{12}/P_{11}$ as these values are closely related to the differences between sequential measurements at different polarizations. A large particle that is present

at a given location in one image, but not present in the corresponding adjacent image will produce a significant artifact. The effect is also evident at low scattering angles, where larger particles tend to represent a larger portion of total scattering.

## 3.2 Refractive index retrievals

Crystalline particles do not take on water until reaching relative humidities above their deliquescence point, generally around 80% in the case of salts. A range of methods are available for calculating the size of a given salt droplet, after the transformation

to an aqueous state has been made. In this work we choose the parameterization proposed by Petters and Kreidenweis (2007) for its simplicity and because the required $\kappa$ parameters are well known for the salts in question. This method states that $gf_{vol}$, the volume growth factor of a particle, can be estimated as:

$$gf_{vol}(RH) = 1 + \kappa \frac{\mathrm{RH}}{1 - \mathrm{RH}} \qquad (3)$$

where RH is the relative humidity of the air surrounding the droplet and $\kappa$ is a constant that is determined by the composition

of the particle in question.





The dry (crystalline) refractive indices of all three salts studied in this work are well known (Tang, 1996) and the resulting wet refractive index can be calculated from the volume mixing rule:

$$n_{wet}(RH) = \frac{(gf_{vol} - 1)\,n_{H_2O} + n_{dry}}{gf_{vol}} \tag{4}$$

where $n_{H_2O}$ is the refractive index of water, $n_{dry}$ is the refractive index of the dry salt and $n_{wet}$ is the refractive index of the
solution (Nessler et al., 2005). Alternative methods for estimating the refractive index of hygroscopic particles exist, but their deviation from the volume mixing rule is less than 1% for solutions that are made up of more than 50% water (Erlick et al., 2011; Schuster et al., 2009).

The refractive indices predicted from equations 3 and 4 are compared with the corresponding GRASP retrievals in Table 1. The ranges of $\kappa$ values given for sodium chloride and ammonium sulfate are taken from Table 3 of Koehler et al. (2006) and
were derived from hygroscopic growth factors in the sub-saturated domain. The $\kappa$ range used for ammonium nitrate are derived from measurements of cloud condensation nuclei (CCN), at super-saturations less the 1%, and originate from Svenningsson et al. (2006), with the spread representing an uncertainty of one standard deviation. Growth factor derived $\kappa$ values were not available for ammonium nitrate but the difference between growth factor and CCN derived $\kappa$ values is generally small compared to the uncertainty in $\kappa$ resulting from measurement errors (Petters and Kreidenweis, 2007). The range in the final
predicted wet refractive indices results from the bounds on the $\kappa$ values, as well as a 2% uncertainty in the RH measurement made inside the PI-Neph.

The retrieved refractive index values are in good agreement with the range predicted by $\kappa$-Köhler theory and the existing literature. Sensitivity studies, performed on ensembles of synthetic data perturbed with modeled PI-Neph noise, suggest one standard deviation uncertainties in retrieved real refractive indices of around 0.02 for non-absorbing particles in the size range
of these humidified salts. These studies also showed a general trend of increasing accuracy in the retrieved real part of the refractive index as the median radius of the particles increased. The converse was true for absorption, where more absorbing particles tended to produces more error in the real refractive index inversion. The agreement between the retrieved and predicted refractive index values is consistent with this error analysis.

The retrieved imaginary parts of the refractive index (not shown) of the ammonium nitrate and ammonium sulfate solutions
were both found to be on the order of $10^{-3}$. These values are indicative of moderate absorption and are slightly larger than values found in the existing literature, which suggests very little absorption ($k < 10^{-7}$) for all three of the solutions measured (Fenn et al., 1985; Toon and Pollack, 1976; Hale and Querry, 1973). An even higher imaginary part of the refractive index ($k = 0.026$) was retrieved in the case of the sodium chloride sample. The magnitude of this value may be, at least in part, related to an unrealistically high retrieved real refractive index. This hypothesis is supported by the fact that constraining the
retrieved real refractive index to the range predicted by the sample RH and $\kappa$-Köhler theory resulted in significantly lower retrieved values of sodium chloride absorption. A comparison was also made between the retrieved single scattering albedo (SSA) and the SSA derived from Particle Soot/Absorption Photometer (PSAP, Radiance Research, Seattle, WA, USA) and integrated scattering measurements (Integrating Nephelometer 3563, TSI Inc., St. Paul, MN, USA) in SEAC[4]RS. A statistically





significant correlation between the two data sets was determined to exist, but the retrieved SSA was also found to systemically overestimate the measured absorption. Notice that the retrieval was based only on scattering measurements (no absorption or extinction data was included) and therefore is expected to show limited sensitivity to these variables. A detailed analysis of the sensitivity of the GRASP/PI-Neph retrieval to absorption is beyond the scope of this work.

After passing their deliquescence point, crystalline salt particles should transform into saline droplets and become spherical in shape. The GRASP/PI-Neph inversion was able to accurately reproduce this spherical morphology in the sodium chloride and ammonium sulfate case, but a spherical fraction of only 54% was retrieved for the ammonium nitrate sample. This deviation from expectation is likely driven by a combination of random error in the PI-Neph measurement and the fact that the scattering of non-spherical particles tends to deviate less from that of spherical particles as particle size decreases. This notion is confirmed

in the sensitivity studies previously described, where it was found that there was very little sensitivity to sphericity in the case of small particles ($r <$200nm).

Retrievals of the monodisperse PSL spheres produced real refractive index values that were within the range of existing measurements available in the literature at all three wavelengths (Bateman et al., 1959; Ma et al., 2003; Sultanova et al., 2003; Jones et al., 2013). The spectral dependence of the retrieved values, as well as the three most recently reported Cauchy Equation

parameterizations of PSL refractive index can be found in Figure 7. The retrieved imaginary part of the refractive index for these spheres was on the order of $10^{-3}$ for all three wavelengths, slightly higher than the previously reported values of around $4 \times 10^{-4}$ (Bateman et al., 1959; Ma et al., 2003).

Figure 8 shows the spectrally dependent distribution of the retrieved dry refractive indices for the 50 chosen SEAC[4]RS cases. The mean retrieved real part of the refractive index at 532nm for the 50 cases, composed primarily of biomass burning

and urban-biogenic mixtures, was found to be 1.53. This figure is in line with the existing measurements made under similar conditions (Shingler et al., 2016), but unfortunately very few airborne, in situ measurements of refractive index are available. Remote sensing retrievals of biomass burning aerosol generally range from 1.47 to 1.55 (Dubovik et al., 2002; Li and Mao, 1990; Westphal and Toon, 1991; Yamasoe et al., 1998), while remote retrievals of urban pollution have generally yielded somewhat lower values, ranging from 1.39 to 1.46 (Dubovik et al., 2002; Redemann et al., 2000). These lower values observed

in the urban pollution remote sensing retrievals are likely driven in large part by particle hygroscopicity. The PI-Neph/GRASP retrievals of real refractive index are expected to be significantly higher in analogous cases as the PI-Neph measurements were made at very low relative humidities, where hygroscopic growth is virtually non-existent. In spite of these differences in measurement conditions, as well as in the sample regions in question, the values are remarkably similar, especially in the case of biomass burning emissions, where hygroscopic influences are expected to be much more limited. Additionally, the spectral

dependence is in line with expectation, and closely matches measurements of common natural aerosol constituents made by Hale and Querry (1973) (Hale and Querry, 1973).

Table 2 shows details of the retrievals performed on the three cases studies. The retrieved real refractive index of the August 19[th] biomass burning plume is slightly higher than the values reported in the literature, and represents the upper end of the values retrieved in the 50 selected samples. The other two cases also returned higher than average values, although they were

more in line with the other samples and typical values reported in the existing literature. The biomass burning particles were





also found to be less absorbing than that of typical smoke, but the values produced by GRASP are in good agreement with direct SSA measurements on aboard the DC-8 derived from PSAP and integrating nephelometer measurements (SEAC4RS, 2015). A significant percentage of particles were determined to be non-spherical in these cases, especially the August 19th biomass burning plume and August 30th forested boundary layer aerosols. The cases on August 19th and August 27th are dominated by

small particles, and in turn there are large uncertainties in the sphericity product. The low spherical percentage retrieved for the August 30th case is potentially realistic given the significance of the coarse mode, but additional independent measurements of sphericity are limited.

### 3.3   Size distribution retrievals

The size distribution retrieved for the PSL spheres is shown in the sub-panel of Figure 7 and agrees well with the manu-
facturer's specifications. The median diameter of the retrieved distribution was found to be 902.7nm which shows excellent agreement with the manufacturer's NIST traceable specification of 903nm±12. It is the authors' experience, based on PI-Neph measurement inversions on a range of PSL products from the same manufacturer, that the uncertainty listed often significantly overestimates the true uncertainty in the central diameter of the size distribution. The full width, 67 percentile (FW67) of the distribution was found to be 17nm, slightly wider than the manufacturers specified FW67 of 8.2nm. Similarly accurate results
sizing PSL spheres with PI-Neph data are demonstrated in Dolgos and Martins (2014) through the use of a Mie theory lookup table.

The retrieved size distributions for all three SEAC$^4$RS case studies are plotted alongside measurements made by dedicated particle sizers in Figure 9. The APS data was converted from aerodynamic to geometric size using an assumed density of $1.3\mathrm{gcm}^{-3}$ and a shape factor of unity. Uncertainties in these assumption can generate significant changes in the resulting
geometric PSD, but the presence of APS data can still be used as an optically independent, qualitative confirmation regarding the presence of significant coarse mode. The UHSAS data is shown for two different calibration aerosols, PSL spheres and ammonium sulfate, which have refractive indices of 1.61 (Jones et al., 2013) and 1.53 (Tang, 1996) respectively. The LAS data shown corresponds to calibration with PSL spheres.

In all three of these cases the peak of the fine mode generally occurs around a radius of 150nm. These values are typical of
the majority of the 50 selected periods, all of which have fine mode median radii (in volume) between 100nm and 200nm. The PI-Neph/GRASP PSD retrievals fall between the two different UHSAS calibrations in each of the three cases, which again is typical of almost all 50 samples.

Among the 50 selected periods for which size distribution comparisons were made, only two cases had coarse modes with volume concentrations that made up a significant portion of the total particle volume. The first of these cases, a sample domi-
nated by transported Saharan dust, had very low aerosol loading and the bulk of the scattering matrix data at scattering angles above $40°$ was below the PI-Neph's limit of detection. The second of these cases, the forested boundary layer measurements taken on August 30th, was therefore chosen as one of the three highlighted case studies. In both cases the size distributions agree remarkably well in the coarse mode, suggesting significant sensitivity to larger particles in the retrieved product. This sensitivity likely resulted primarily from the PI-Neph's ability to measure down to scattering angles as low as $3°$ during SEAC$^4$RS.



Lienert et al. (2003) was also able to show sensitivity to super-micron particles given a minimum scattering angle of around $2°$. On the other hand, Sviridenkov et al. (2014) determined that single scattering measurements over a scattering angle range of $10°$ to $90°$, were insufficient to provide significant information about the coarse mode. All of these conclusions are in agreement with theoretical sensitivity studies indicating that measurements at very low scattering angles are required if the coarse mode is to be accurately recovered (Dubovik et al., 2000).

In order to simplify the comparison of the retrieved size distributions with those measured by the dedicated aerosol spectrometers, the fine mode of each PSD was parameterized according to three metrics: total volume concentration, median radius and the span of the distribution. When determining these metrics the values of the volume distributions corresponding to radii less than 50nm were first removed, as this lower bound corresponds to the bottom of the PI-Neph/GRASP retrieval range. The upper end of the remaining size distribution was then further truncated to include only fine mode particles. The division between the fine and coarse modes was defined as the minimum value of the LAS volume distribution, closest to r=300nm. A visual inspection of all cases confirmed that this metric was sufficient to reasonably isolate the fine mode when two modes were present. The volume concentration, median ($r_{50}$) and span ($(r_{90} - r_{10})/r_{50}$) were then calculated using theses final truncated volume distributions. Linear interpolation was used when the $10^{th}$, $50^{th}$ or $90^{th}$ percentile values, as well as the bounds of the truncated distributions, fell between the midpoints of two size bins. Scatter plots showing the results of these parameterizations for the three OPC measurements vs the corresponding PI-Neph retrieval are shown in Figure 10.

The PI-Neph retrieved volume concentrations and median radii generally fall somewhere between the two different UHSAS calibrations, with the best agreement generally tending towards the ammonium sulfate calibration. This is consistent with the average retrieved refractive index for the 50 cases (n=1.53) which is in very close agreement to the dry refractive index of ammonium sulfate found in the literature. The LAS consistently measured smaller and fewer particles than all the other sizing techniques, but still showed significant correlation with the PI-Neph/GRASP retrievals. There was weaker agreement regarding the width of the distribution among the four techniques. The retrieved spans generally best matched the corresponding PSL calibrated UHSAS values, but the inverted values covered a larger range of spans than the values measured by the OPCs. The PI-Neph retrieved spans fell between 0.55 and 1.03 in 95% of the cases. In contrast, the LAS showed the least variability in span, with 95% of the values falling between 0.65 and 0.85. The differences in span between PI-Neph retrievals and the OPCs was likely driven in large part by their different sampling techniques (ensemble vs single particle measurements).

The large differences between UHSAS measurements under different calibrations, with disparate refractive indices, demonstrates the significance of the refractive index assumptions required. The results of this work, as well as others (Shingler et al., 2016), suggest that the real refractive indices of natural aerosol can frequently reach values as low as 1.48 at 532nm. This is substantially lower than the refractive index of ammonium sulfate (n = 1.53), which has the lowest value of the aerosols that are commonly used to calibrate optical particle sizers, and further emphasizes the significance of the basis resulting from uncertainty in refractive indices.

In order to further asses the retrieval variability, resulting from changes in refractive index and sphericity, the 50 SEAC[4]RS cases were inverted a second time with assumptions corresponding to PSL spheres. In this analysis the refractive index was forced match measurements of PSL and non-spherical particles were excluded from GRASP's aerosol model. This configura-



tion produced significantly better agreement with the PSL calibrated UHSAS measurements in volume concentration, median radius and span, when compared to the unconstrained retrievals. This result further demonstrates that differences in fundamental assumptions about the optical and morphological properties of the particles are driving a significant portion of the differences between the retrieved and measured values.

## 4   Conclusions

This work represents the first time that aerosol optical and microphysical properties were retrieved from airborne, polar nephelometer data. Additionally, the PI-Neph/GRASP inversion makes fewer assumptions regarding the shape of the recovered size distribution and particle sphericity, then previous in situ light scattering retrievals. The resulting products are in good agreement with expectations, and compare well with existing measurement techniques. Furthermore, the GRASP fit to PI-Neph data is consistent with the PI-Neph's level of error, indicating that the assumptions made in the retrieval are sufficient to faithfully reproduce the light scattering of realistic, ambient aerosols.

The real refractive index of humidified salts retrieved with this method agree well with the predictions made by $\kappa$-Köhler theory and existing dry measurements. The PI-Neph retrieval of PSL refractive index agrees with other contemporary techniques to within the deviation present in those reported values. Furthermore, inversions of airborne SEAC[4]RS data produced refractive indices that were in good agreement with the existing literature.

There is significant spread in the aerosol size distribution measurements made by the OPCs, but the corresponding PI-Neph/GRASP retrievals generally fall within the range of the existing measurements. A major part of the differences in the measured size distribution from these instruments stem from the need to assume a refractive index during the calibration process. The PI-Neph/GRASP retrieval has sufficient sensitivity to constrain the refractive index with enough accuracy to potentially reduce these biases. The fact that the PSD retrievals fell between the two UHSAS calibrations, in a manner consistent with the retrieved refractive index, supports this conclusion.

The PI-Neph inversions have also shown moderate sensitivity to absorption but a detailed assessment of the accuracy of this retrieved parameter is beyond the scope of this paper and will have to remain the subject of future study. Additionally, promising results were obtained regarding the retrieval of sphericity in the case of the humidified salts as well as in sensitivity studies, but as a result of the limited morphological information available in the SEAC[4]RS dataset, a robust evaluation of this product is limited at this time.

## 5   Code availability

The GRASP software package is open-source and available for download by request at http://www.grasp-open.com.





## 6  Data availability

All relevant measurements made during the SEAC[4]RS experiment are available through the SEAC[4]RS data archive at http://www-air.larc.nasa.gov/missions/seac4rs/ (SEAC4RS, 2015). Requests for additional data can be made to the corresponding author at reedespinosa@umbc.edu.

5   *Acknowledgements.*  We acknowledge funding support from the NASA Earth Science Enterprise for the SEAC[4]RS campaign under Grant NNX12AC37G, and under the Atmospheric Composition Campaign Data Analysis and Modeling Program (ACCDAM) grant NNX14AP73G, both managed by Dr. Hal Maring. The authors would also like to thank the members of the LARGE group, particularly Bruce Anderson, Edward Winstead and Lee Thornhill for their support incorporating the PI-Neph into the LARGE instrument package. We are also grateful for the scientific and technical support of the LACO team at UMBC, especially Dominik Cieslak and Frank Harris. Additionally, we would
10  like to thank the entire SEAC[4]RS science team for providing supporting data and relevant discussion.



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





**Table 1.** Predicted and retrieved refractive indices, median radii in volume and spherical fractions for the three artificially generated aerosols. Also shown are the deliquescence relative humidities (DRH), $\kappa$ values, and dry refractive indices taken from the literature. All refractive indices are at 532nm.

| Compound | DRH(%) | Measured RH(%) | $\kappa$ | $r_{50}^{GRASP}$(nm) | $Sphere(\%)$ | $n_{dry}$ | $n_{wet}^{GRASP}$ | $n_{wet}^{\kappa\text{Köhler}}$ |
|---|---|---|---|---|---|---|---|---|
| $NaCl$ | 80 | $83.7\pm2$ | 0.91-1.33 | 144 | 100 | 1.544 | 1.395 | 1.353-1.372 |
| $(NH_4)_2SO_4$ | 75 | $82.6\pm2$ | 0.33-0.72 | 120 | 100 | 1.530 | 1.383 | 1.370-1.414 |
| $NH_4NO_3$ | 62 | $83.5\pm2$ | 0.58-0.75 | 129 | 54 | 1.554 | 1.392 | 1.371-1.393 |

**Table 2.** PI-Neph derived total scattering as well as retrieved complex refractive index, sphere fraction and SSA for the three highlighted case studies. Additionally, the SSA derived from PSAP and integrating nephelometer measurements is shown for comparison. All spectrally dependent parameters are listed at 532nm.

| Aerosol Case | Date | Time(UTC) | $\beta_{scat}$ | $m_{GRASP}$ | $Sphere_{GRASP}$ | $SSA_{GRASP}$ | $SSA_{PSAP}$ |
|---|---|---|---|---|---|---|---|
| BB Plume #1 | Aug 19[th] | 19:06-19:13 | $489\text{Mm}^{-1}$ | $1.594+0.005i$ | 64.5% | 0.976 | 0.964 |
| BB Plume #2 | Aug 27[th] | 21:42-21:48 | $95.9\text{Mm}^{-1}$ | $1.565+0.007i$ | 91.0% | 0.962 | 0.959 |
| Forrested BL | Aug 30[th] | 20:55-21:12 | $41.9\text{Mm}^{-1}$ | $1.569+0.009i$ | 48.9% | 0.932 | 0.930 |

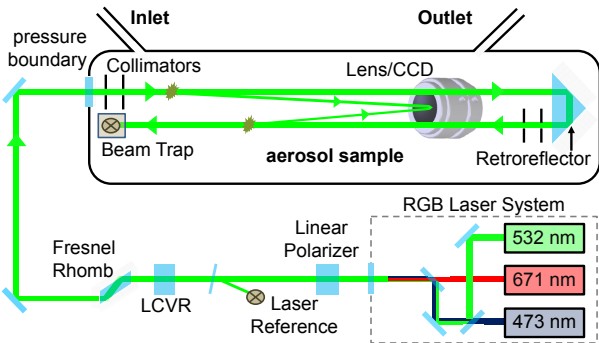

**Figure 1.** The PI-Neph instrument concept. Figure adapted from Dolgos and Martins (2014).





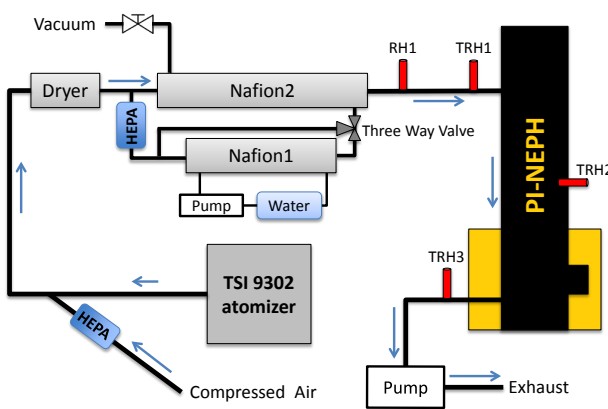

**Figure 2.** Laboratory aerosol generation instrumental setup used to suspend salts and PSL spheres.

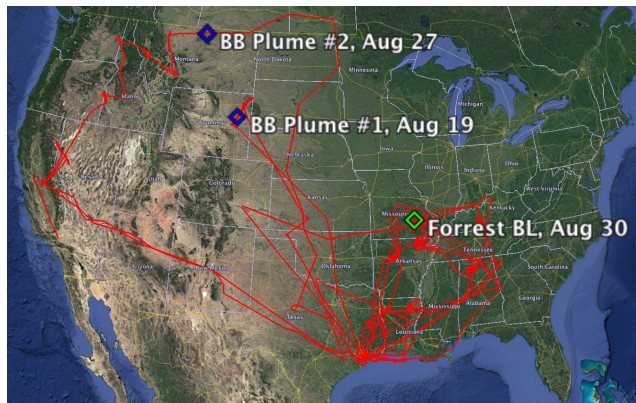

**Figure 3.** Flight paths of the ten SEAC[4]RS flights from which data is used in this paper. Additionally, three specific case studies are called out with diamonds. The case studies include two biomass burning dominated aerosols (blue) as well as measurements made in the boundary layer of a forested region in south east Missouri (green).





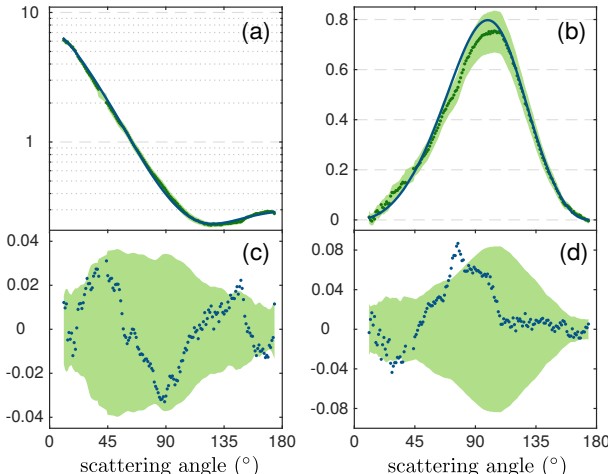

**Figure 4.** PI-Neph measurements at 532nm (points) with $2\sigma$ instrumental error (green fill) and the GRASP retrieval best fit (solid line) for ammonium sulfate measurements made in the laboratory. Panel (a) shows normalized $P_{11}$ data plotted on a log scale, while panel (b) shows $-P_{12}/P_{11}$ data on a linear scale. Panel (c) shows the $P_{11}$ differences according to the log transformation described in equation 1, while the conventional residuals in $-P_{12}/P_{11}$, as given by equation 2, are plotted in (d).

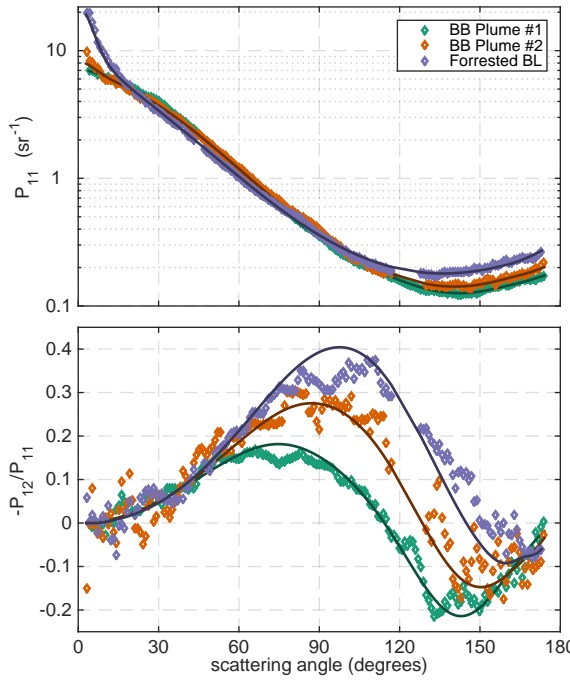

**Figure 5.** Scattering matrix elements (diamonds) measured by the PI-Neph at 532nm and the corresponding GRASP fits (solid lines) for the three highlighted SEAC[4]RS aerosol samples.





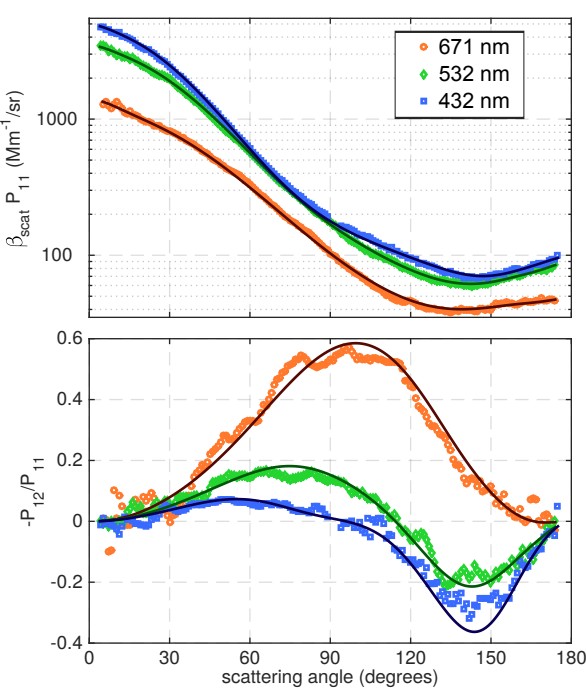

**Figure 6.** Scattering matrix elements at 473nm (blue), 532nm (green) and 671nm (red) measured in BB plume #1 on August 19[th] along with the corresponding GRASP fits (solid lines).





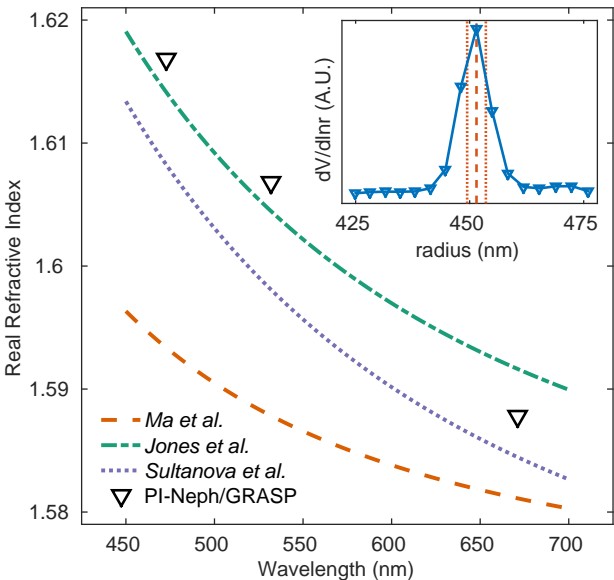

**Figure 7.** Retrieved refractive index of PSL spheres, alongside three previous, modern measurements of polystyrene refractive indices (Ma et al., 2003; Jones et al., 2013; Sultanova et al., 2003). The subplot shows the retrieved size distribution (blue) along side the manufacture's specified central radius (red dashes) and FW67 (red dots).

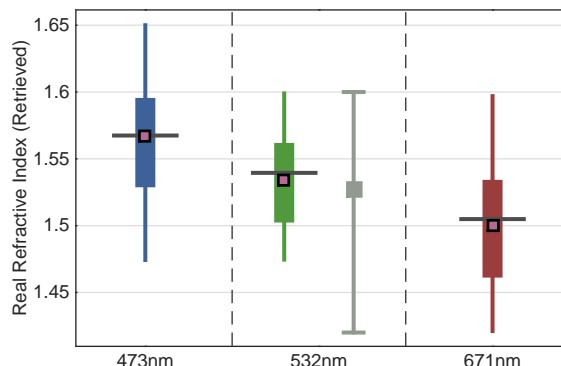

**Figure 8.** Retrieved refractive index at all three PI-Neph wavelengths for the 50 selected SEAC[4]RS samples. Box and whiskers plots show the data distribution by quartile while the connected black squares show the spectral dependence of the mean. The gray bounds at 532nm denote the minimum and maximum values measured by Shingler et al. (2016) in SEAC[4]RS while the grey square denotes the corresponding mean.





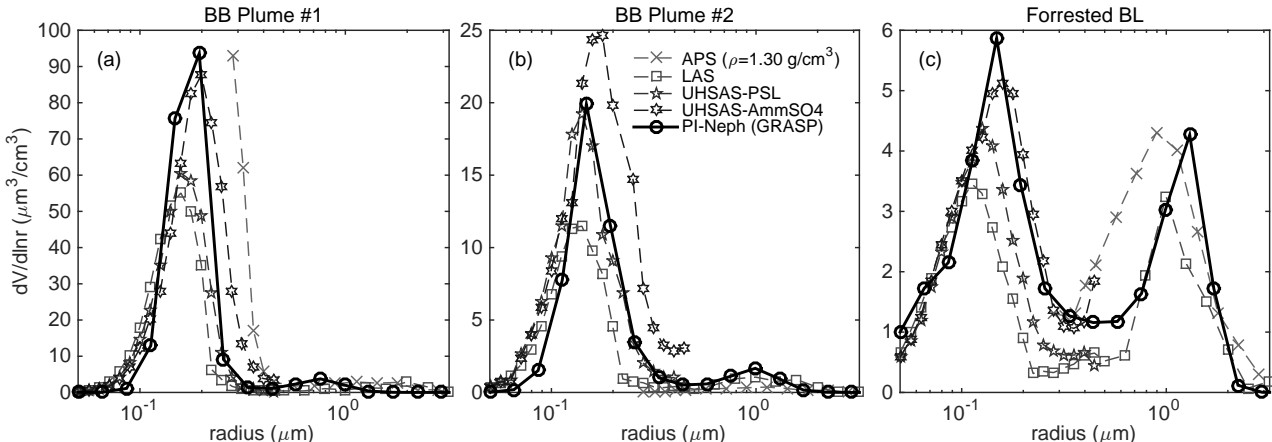

**Figure 9.** Direct comparisons of PI-Neph/GRASP retrieved size distributions with dedicated particle sizers that sampled in parallel to the PI-Neph. The three cases selected show measurements from the (a) August 19th and (b) August 27th biomass burning cases, as well as (c) boundary layer measurements made above a forested region of south east Missouri on August 30th.

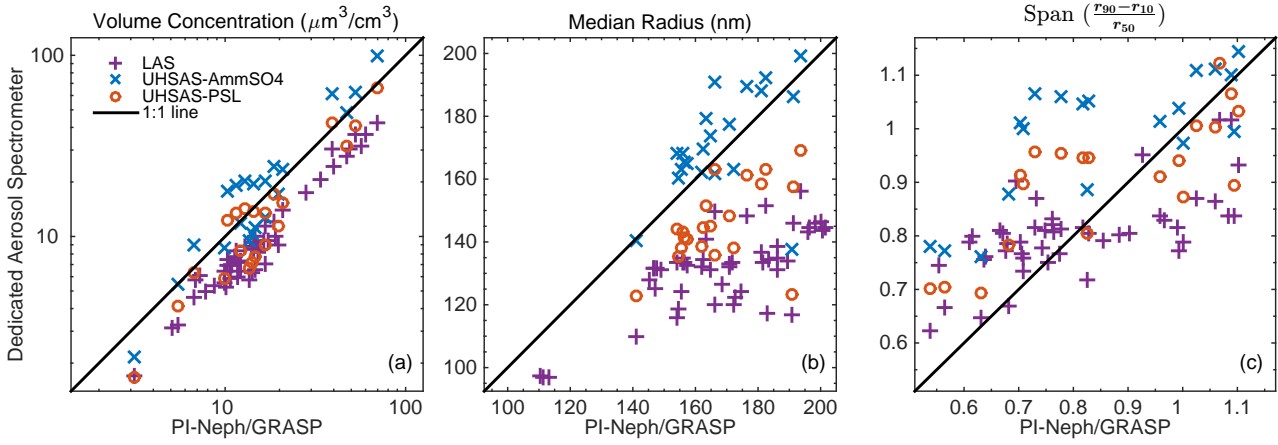

**Figure 10.** Scatter plot comparisons of retrieved size distributions with particle sizers sampling in parallel to the PI-Neph. In order from left to right the panels show total fine mode (a) volume concentration, (b) volume median radius and (c) $span = \frac{r_{90} - r_{10}}{r_{50}}$. The value retrieved from PI-Neph measurements is plotted on the horizontal axis while the value measured by the corresponding dedicated aerosol spectrometer is plotted along the vertical axis. The comparisons are made against LAS measurements (purple pluses), UHSAS ammonium-sulfate equivalent optical diameters (blue crosses) and UHSAS PSL equivalent optical diameters (red circles).