# Peer review of "Retrievals of Aerosol Optical and Microphysical Properties from Imaging Polar Nephelometer Scattering Measurements"

_Atmospheric Measurement Techniques, 2016_

## Referee Comment (RC1) · Anonymous Referee #1 · 5 Dec 2016

General comments:

The inversion algorithm GRASP is applied to airborne and laboratory measurements made with the novel Polarized Imaging Nephelometer PI-Neph. The final goal is to retrieve airborne aerosol size distributions, complex refractive indices, and percentage of spherical particles. The airborne measurements were made aboard the NASA DC-8 aircraft during the SEAC[4]RS experiment. The selected set of data contains the highest aerosol scattering levels during fifty separate sampling periods over the course of ten different flights. It represents a wide range of aerosols including urban pollution, organics, Saharan dust, and biomass burning. Simultaneous measurements with other two optical and one aerodynamic particle sizers aboard the aircraft are used for

inter-comparison with PI-Neph retrievals.

The combination of the complex inversion algorithm GRASP with the PI-Neph airborne measurements of the $P_{11}$ and $-P_{12}/P_{11}$ elements of the scattering matrix at three different wavelengths mark a significant step forward in retrieving physical properties of atmospheric aerosols. The manuscript provides a detailed description of the experimental approach and methodology in the retrieval procedure. Moreover, it gives a quite critical presentation of their own results which is highly laudable. Therefore, I strongly recommend publication of this paper on Atmospheric Measurements Techniques. However, there are some minor issues that I would like the authors to clarify before publication.

Specific comments

- Page 3, lines 19-20: "This compact device (PI-Neph) is capable of measuring scattering matrix elements with an angular resolution and range that has previously been unavailable in polar nephelometers." As mentioned in page 4, lines 5-6, "PI-Neph measurements are generally made over and angular range of 3 to 177 degrees in zenith scattering angle, with an angle resolution of less than one degree".

The authors do not seem to be aware of the polar nephelometer that is in operation for more than six years covering the scattering angle range from 3 to 177 degrees with an angular resolution of 1/8 degrees (Muñoz et al. Icarus, 211, 894-900, 2011).

- Page 4, line 4: $\beta_{sca}$ must be defined.

- Page 4, line 9. ".. the nearest neighbor technique used to extrapolate the $P_{11}$ at the truncated regions around 0 and 180 degrees".

A meaningful extrapolation of the experimental diffraction forward peak (0-3 degrees) is not trivial see e.g. Liu et al. JQSRT 79-80, 911-920, 2003. Please, include the description of the extrapolation procedure.

- Figure 6 (top panel) presents the spectral dependence of the absolute phase functions

in BB plume1 together with the GRASP fits. How are the absolute phase functions obtained out of the measured values?

- Page 9, lines 25-30. It is claimed that the overestimated imaginary part of the refractive index for the ammonium nitrate, ammonium sulfate, and sodium chloride may be due to an unrealistically high retrieved real refractive index. The imaginary part of the refractive index is also overestimated in the Polystyrene spheres case reported in page 10, lines 12-16, even when the retrieved real part of the refractive index is in agreement with the nominal values. That seems to indicate that GRASP could produce a systematic overestimation of the imaginary part of the refractive index. As claimed in the paper "a detailed analysis of the sensitivity of the GRASP/Pi-Neph retrieval to absorption is beyond the scope of this paper". I agree that a more detailed study is needed and it is beyond the scope of this paper. In any case as a future work I recommend the work of Zubko et al. JQSRT, 151, 38-42, 2015, where the location and amplitude of the negative polarization minimum is used to retrieve the complex refractive index for dust grains.

- Page 10. A figure with the spectral dependence of the measured $P_{11}$ and $-P_{12}/P_{11}$ elements and corresponding fits for the monodisperse PSL spheres must be included.

- Figure 6 proves the accuracy of GRASP in reproducing the spectral dependence of the $P_{11}$ and $-P_{12}/P_{11}$ elements in the case of fine mode biomass particles i.e. BB plume1. A similar figure must be included for the Forested BL case that presents a size distribution with coarse mode. That would be an important test for the performance of GRASP and the spheroidal model. As mentioned in the paper, the scattering of non-spherical particles tends to deviate less from that of spherical particles as particle size decreases.
* * *

---

## Referee Comment (RC2) · Anonymous Referee #2 · 22 Jan 2017

In the paper laboratory and field measurements of the angular light scattering and polarization of aerosol particles are used to retrieve the aerosol microphysical properties and the complex refractive index. For the retrieval the comprehensive GRASP software package was used. This approach is novel and therefore within the scope of AMT. The paper is well structured and the results are clearly presented. I therefore recommend the publication in AMT after the following major issue has been addressed.

Section 3.2 has to be carefully revised as for me it is not convincing that any reasonable value for the imaginary part of the refractive index can be given based on the presented method. Including a measurement that is sensitive to the particle absorption is essential to retrieve the imaginary part of the refractive index, especially in case of weakly

absorbing aerosols. Although the authors are aware of this problem - as it is indicated by two sentences in Sec. 3.2 and in the Conclusions - they give imaginary parts of the refractive index for the case studies in Table 2. This is misleading to the reader, who might take these values as approved in his own work. I recommend to remove any values of the imaginary part of the refractive index from the paper

---

## Author Comment (AC1) · 2 Feb 2017

**Responses to Anonymous Referee #1**

We would like to thank the reviewer for his/her time, thoughtful insights and helpful comments. A point-by-point response to each of the reviewers concerns is listed below. The reviewers comments are shown in bold italics, while the authors' responses are indented and displayed in regular type.

***Page 3, lines 19-20: "This compact device (PI-Neph) is capable of measuring scattering matrix elements with an angular resolution and range that has previously been unavailable in polar nephelometers." As mentioned in page 4, lines 5-6, "PI-Neph measurements are generally made over and angular range of 3 to 177 degrees in zenith scattering angle, with an angle resolution of less than one degree".***
***The authors do not seem to be aware of the polar nephelometer that is in operation for more than six years covering the scattering angle range from 3 to 177 degrees with an angular resolution of 1/8 degrees (Munoz et al. Icarus, 211, 894-900, 2011).***

> We appreciate the clarification on the capabilities of the IAA light scattering instrument. The sentence on Page 3, around lines 19-20 has been updated to the following:
> "This setup permits the construction of an instrument that is compact and stable enough to be flown on a variety of airborne platforms, while still allowing for measurements of scattering matrix elements over an angular resolution and range that is comparable to state of the art laboratory techniques (Muñoz et al., 2011)."

***Page 4, line 4: $\beta_{sca}$ must be defined.***

> In an effort to improve clarity, the prior use of the notation $\beta_{sca}P_{ij}$ to represented absolute scattering matrix elements has been replaced by $F_{ij}$. $\beta_{sca}$ is still used in Table 2 to represent the total integrated scattering coefficient and an explanation of the variable has been added to the corresponding caption:
> " Truncation corrected total scattering ($\beta_{sca}$) from the integrating nephelometer..."

***Page 4, line 9. ".. the nearest neighbor technique used to extrapolate the $P_{11}$ at the truncated regions around 0 and 180 degrees". A meaningful extrapolation of the experimental diffraction forward peak (0-3 degrees) is not trivial see e.g. Liu et al. JQSRT 79-80, 911-920, 2003. Please, include the description of the extrapolation procedure.***

> We acknowledge the potential bias in the normalization produced by our relatively crude extrapolation technique and we have chosen to re-normalize the phase functions Figure 5 such that $\tilde{F}_{11}(30°) = 1$. Correspondingly, the sentence on Page 4, line 9 has been changed to the following:
> "Additionally, normalized phase functions are represented by $\tilde{F}_{11}$ in this paper and are scaled such that $\tilde{F}_{11}(30°) = 1$."
> The phase function shown in Figure 4 has been changed to absolute units to minimize the complications associated with displaying errors when the measurement is scaled according to the data from a single angle.

[Figure]

Figure 4: PI-Neph measurements at 532nm (points) with $2\sigma$ instrumental error (gray fill) and the GRASP retrieval best fit (solid line) for ammonium sulfate measurements made in the laboratory. Panel (a) shows absolute $F_{11}$ $(Mm^{-1}/sr)$ data plotted on a log scale, while panel (b) shows $-F_{12}/F_{11}$ data on a linear scale. Panel (c) shows the $F_{11}$ differences according to the log transformation described in equation 1, while the conventional residuals in $-F_{12}/F_{11}$, as given by equation 2, are plotted in (d).

[Figure]

Figure 5: Normalized scattering matrix elements (circles) measured by the PI-Neph at 532nm and the corresponding GRASP fits (solid lines) for the three highlighted SEAC[4]RS aerosol samples.

*Figure 6 (top panel) presents the spectral dependence of the absolute phase functions in BB plume1 together with the GRASP fits. How are the absolute phase functions obtained out of the measured values?*

The PI-Neph makes direct measurements of absolute phase function. The following text was added to the third paragraph of section "2.1 Polarized Imaging Nephelometer" to clarify this point:

"The images can then be processed in a manner that allows for direct measurements of the absolute phase function $F_{11}(\theta)$ and $F_{12}(\theta)$, with $\theta$ representing the zenith scattering angle (azimuthal symmetry is implied by the assumption of a macroscopically isotropic and symmetric medium). Measurements of molecular scatterers ($CO_2$ and $N_2$), whose absolute scattering matrix elements are well characterized (Anderson et al., 1996; Young, 1980), allow for the determination of unique calibration constants for each angle. This angular dependent absolute calibration allows for direct measurements of absolute phase function in known units ($Mm^{-1}/sr$), free from any truncation error."

***Page 9, lines 25-30. It is claimed that the overestimated imaginary part of the refractive index for the ammonium nitrate, ammonium sulfate, and sodium chloride may be due to an unrealistically high retrieved real refractive index. The imaginary part of the refractive index is also overestimated in the Polystyrene spheres case reported in page 10, lines 12-16, even when the retrieved real part of the refractive index is in agreement with the nominal values. That seems to indicate that GRASP could produce a systematic over-estimation of the imaginary part of the refractive index. As claimed in the paper "a detailed analysis of the sensitivity of the GRASP/Pi-Neph retrieval to absorption is beyond the scope of this paper". I agree that a more detailed study is needed and it is beyond the scope of this paper. In any case as a future work I recommend the work of Zubko et al. JQSRT, 151, 38-42, 2015, where the location and amplitude of the negative polarization minimum is used to retrieve the complex refractive index for dust grains.***

The authors appreciate the insights and reference provided.

***Page 10. A figure with the spectral dependence of the measured $P_{11}$ and $-P_{12}/P_{11}$ elements and corresponding fits for the monodisperse PSL spheres must be included.***

A figure showing the measurements and fits for the PSL case (below) has been included. Additionally, the following paragraph was added to the end of section 3.1:

"The monodisperse PSL measurements and corresponding GRASP fits (shown in Figure 8) agree well in the case of $F_{11}$. Overall there is also good agreement in the $-F_{12}/F_{11}$ data, but some significant deviations do occur. The GRASP size distribution retrieval for this case had a full width, 67 percentile (FW67) of 17nm, which is more than twice the width specified by the manufacturer (FW67=8.2nm). However, a narrower size distribution corresponding to the manufacturer's specification was found to reproduce some features of the measurement significantly better than GRASP's original retrieval. This improvement was most apparent in the 473nm and 532nm $-F_{12}/F_{11}$ data, particularly at scattering angles between 20° and 60° where Mie theory predicts $-F_{12}/F_{11}$ to have high sensitivity to the distribution's width. Further studies indicated that GRASP was able to reproduce $-F_{12}/F_{11}$ corresponding to this narrower PSD with high accuracy when noise free synthetic data was used as input. Additionally, running retrievals on the measured data using increasingly finer size resolution kernels did not improve the retrieval's ability to fit these features. The deviations in the fit were thus determined to be the result of GRASP's sensitivity to certain characteristics of the noise in the

measured data, not insufficient size resolution in the fine resolution kernels used in the PSL case."

[Figure]

Figure 8: Scattering matrix elements at 473nm (blue), 532nm (green) and 671nm (red) for 903nm diameter PSL sample along with the corresponding GRASP fits (solid lines).

***Figure 6 proves the accuracy of GRASP in reproducing the spectral dependence of the $P_{11}$ and $-P_{12}/P_{11}$ elements in the case of fine mode biomass particles i.e. BB plume1. A similar figure must be included for the Forested BL case that presents a size distribution with coarse mode. That would be an important test for the performance of GRASP and the spheroidal model. As mentioned in the paper, the scattering of non- spherical particles tends to deviate less from that of spherical particles as particle size decreases.***

A figure showing the $P_{11}$ and $-P_{12}/P_{11}$ elements from the Forested BL case (below) has been included in the text. The following two sentences discussing the figure were also added to the end of the paragraph beginning on line 9 of page 8:

"The same variables are plotted for the the forested boundary layer case in Figure 7 to show the spectral dependence of the measured scattering matrix elements and the corresponding fits when a significant coarse mode is present. In this last case, low aerosol concentrations and greater than average straylight levels inside the instrument resulted in a gap in the 473nm $F_{12}$ measurements between 80° and 142° in scattering angle."

[Figure]

Figure 7: Scattering matrix elements at 473nm (blue), 532nm (green) and 671nm (red) measured over a forested region of southeast Missouri along with the corresponding GRASP fits (solid lines).

**References**

T. L. Anderson, D. S. Covert, S. F. Marshall, M. L. Laucks, R. J. Charlson, A. P. Waggoner, J. A. Ogren, R. Caldow, R. L. Holm, F. R. Quant, G. J. Sem, A Wiedensohler, N. A. Ahlquist, and T. S. Bates. Performance Characteristics of a High-Sensitivity, Three-Wavelength Total Scattering/Backscatter Nephelometer. *Journal of Atmospheric and Oceanic Technology*, 13:967–986, 1996.

O. Muñoz, F. Moreno, D. Guirado, J. L. Ramos, H. Volten, and J. W. Hovenier. The IAA cosmic dust laboratory: Experimental scattering matrices of clay particles. *Icarus*, 211(1): 894–900, 2011. ISSN 00191035. doi: 10.1016/j.icarus.2010.10.027.

Andrew T Young. Revised depolarization corrections for atmospheric extinction. *Applied optics*, 19(20):3427–3428, 1980.

---

## Author Comment (AC2) · 2 Feb 2017

The comment was uploaded in the form of a supplement:
http://www.atmos-meas-tech-discuss.net/amt-2016-356/amt-2016-356-AC2-
supplement.pdf
* * *